# Age, sex and primary care setting differences in patients' perception of community healthcare seeking behaviour towards health services

**Ming Tsuey Lim**[1]*, **Yvonne Mei Fong Lim**[1], **Seng Fah Tong**[2], **Sheamini Sivasampu**[1]

**1** Centre for Clinical Outcomes Research, Institute for Clinical Research, National Institutes of Health (NIH), Ministry of Health Malaysia, Jalan Setia Murni U13/52, Seksyen U13, Bandar Setia Alam, Shah Alam, Selangor, Malaysia, **2** Department of Family Medicine, Universiti Kebangsaan Malaysia, UKM, Bangi Selangor, Malaysia

* mtlim2000@yahoo.com

## Abstract

### Introduction

Understanding the potential determinants of community healthcare seeking behaviour helps in improving healthcare utilisation and health outcomes within different populations. This in turn will aid the development of healthcare policies and planning for prevention, early diagnosis and management of health conditions.

### Objective

To evaluate patients' perception of community healthcare seeking behaviour towards both acute and preventive physical and psychosocial health concerns by sex, age and type of primary care setting (as a proxy for affordability of healthcare).

### Methods

A total of 3979 patients from 221 public and 239 private clinics in Malaysia were interviewed between June 2015 and February 2016 using a patient experience survey questionnaire from the Quality and Cost of Primary Care cross-sectional study. Multivariable logistic regression analysis adjusted for the complex survey design was used.

### Results

After adjusting for covariates, more women than men perceived that most people would see their general practitioners for commonly consulted acute and preventive physical and some psychosocial health concerns such as stomach pain (adjusted odds ratio (AOR), 1.64; 95% confidence interval (CI), 1.22–2.21), sprained ankle (AOR, 1.29; 95% CI, 1.06–1.56), anxiety (AOR, 1.32; 95% CI, 1.12–1.55), domestic violence (AOR, 1.35; 95% CI, 1.13–1.62) and relationship problems (AOR, 1.24; 95% CI, 1.02–1.51). There were no significant differences in perceived healthcare seeking behaviour by age groups except for the removal of a

**Data Availability Statement:** All relevant data are provided within the manuscript. Individual level data cannot be made publicly available because

this would potentially compromise patient confidentiality. The data collected within this study is managed in accordance with the Personal Data Protection Act in Malaysia and is subject to ethical restrictions under the Medical Research and Ethics Committee, Ministry of Health Malaysia. Data is available on request and interested researchers may address data access requests to the Institute for Clinical Research at contact@crc.gov.my.

**Funding:** The Malaysian QUALICOPC study is funded by a research grant from the Ministry of Health, Malaysia (NMRR-15-607-25769) under the Malaysian Health Systems Research initiative. The funders had no role in the study design, data collection and analysis, decision to publish, or preparation of the manuscript.

**Competing interests:** The authors have declared that no competing interests exist.

wart (AOR, 1.41; 95% CI, 1.12–1.76). Patients who visited the public clinics had generally higher perception of community healthcare seeking behaviour for both acute and preventive physical and psychosocial health concerns compared to those who went to private clinics.

## Conclusions

Our findings showed that sex and healthcare affordability differences were present in perceived community healthcare seeking behaviour towards primary care services. Also perceived healthcare seeking behaviour were consistently lower for psychosocial health concerns compared to physical health concerns.

## Introduction

Healthcare seeking behaviour involves decision-making or actions taken for any health-related problems at the community or household level [1]. These decisions take into consideration all available healthcare options within public or private sectors as well as formal and informal healthcare services [1]. Studies have shown that various determinants such as sex, age, social status, type of illness, access to services and perceived quality of the service [1, 2] affect an individual's healthcare seeking behaviour but findings have been inconsistent. As such, it is necessary to understand these potential determinants in improving healthcare utilisation and health outcomes within different populations. Information on healthcare seeking behaviour and patterns of healthcare utilisation in turn will aid the development of healthcare policies and planning for prevention, early diagnosis and management of health conditions [3]. In addition, early healthcare utilisation and appropriate interventions allow reduction of healthcare costs, disability and death from diseases [4].

In this study, we focus on perceived community healthcare seeking behaviour in primary care as primary care is often patients' first point of contact with the healthcare system for most medical problems. It offers a wide spectrum of clinical services spanning from prevention, screening, treatment and rehabilitation of all major health conditions, particularly non-communicable diseases [5, 6]. Thus far, patient surveys on healthcare seeking behaviour have often focused on disease specific areas such as tuberculosis [7] or depression [8], while less attention has been given to primary care in general. Understanding this is important because the community perception provides another indicator of primary care utilisation. We aim to bridge this gap in knowledge on healthcare seeking behaviour which will not only reflect the patterns of perceived healthcare seeking behaviour at the different stages of utilisation, access and barriers to healthcare services in a given community but also the potential determinants to community perceptions [9]. Hence, perceived healthcare seeking behaviour can serve as a tool to understand how healthcare services are used and also to gauge future demand for healthcare services.

It was found that age, sex and accessibility or affordability to healthcare are some of the main determinants which significantly influence healthcare seeking behaviour among different population segments [1, 10]. However, reviews showed lack of consistency in the associations between these determinants and healthcare utilization [10, 11]. Few have investigated how people generally perceive the community's intention to seek primary care consultations and it is known that help seeking patterns amongst people are influenced by community norms [2].

We hypothesise that sex, age and health insurance or affordability of care are key determinants for perceived community healthcare seeking behaviour for a range of physical and psychosocial health problems in Malaysia. Therefore, the aim of this study is to determine the perceived community healthcare seeking behaviour for acute and preventive physical and psychosocial health concerns in Malaysian primary care clinics with a focus on differences in age, sex and type of primary setting (as a proxy for affordability of healthcare).

## Materials and methods

### Study design

The data for this study was extracted from the International Quality and Cost of Primary Care (QUALICOPC) study conducted in the primary care setting in Malaysia [12, 13]. Data collection was conducted between July 2015 and February 2016 across five states (Wilayah Persekutuan Kuala Lumpur, Selangor, Sabah, Sarawak, and Kelantan), which were representative of the primary care population in Malaysia in terms of patient demographics and disease patterns. The QUALICOPC study aimed to collect information from patients on their experiences on the coordination, continuity, quality and equity in primary care. A total of 221 and 239 clinics were randomly sampled from the public and private sector respectively. We stratified the clinics by state and geographical location (urban or rural) and sampled clinics proportionately to the size of each stratum. Further details on the methods including study design, sampling frame, sampling methods, questionnaires and eligibility criteria are described elsewhere [12, 13].

### Study population

Ten patients aged 18 years old and above from each clinic were invited to participate in the survey. One patient will be administered the patient values questionnaire while nine others will be administered the patient experience questionnaire [14]. They were informed that participation was voluntary and would not affect the provision of medical care. They were also assured that their responses will be treated with strict confidentiality and no identifiers were collected. A total of 4983 patients were invited and the overall response rate for both patient questionnaires was 91.1%. A non-response analysis was not necessary as it is usually conducted only when response rates fall below 80% [15]. One hundred and three respondents were excluded because of incomplete data. Of the two types of questionnaires administered, we used data from only the patient experience survey in this analysis.

Ethical approval was granted by the Medical Research and Ethics Committee, Ministry of Health Malaysia (NMRR-15-607-25769).

### Survey tool

The QUALICOPC study comprised four sets of questionnaires namely (i) practice, (ii) doctor, (iii) patient experience and (iv) patient values questionnaires [14]. We adapted the original QUALICOPC questionnaire to the local primary care setting but kept as close possible to the original versions. The patient questionnaires were translated into Malay and Chinese languages using the forward-backward translation process [12, 13] as they were sets that answered by general public. Only the sub-section on the community healthcare seeking behaviour perception (12 items) from the patient experience questionnaire was presented in this analysis and all patients who responded to this questionnaire were included.

The outcome measures were based on 8 acute and preventive physical and 4 psychosocial health concerns, which begins with the following question: "Would most people visit a clinic doctor for:". The 12 items were divided into two groups as shown below:

a. Acute and preventive physical health concerns

 i. A cut finger that needs to be stitched

 ii. Removal of a wart

 iii. Routine health checks

 iv. Deteriorated vision

 v. Stomach pain

 vi. Blood in stool

 vii. Sprained ankle

 viii. Help to quit smoking

b. Psychosocial health concerns

 i. Anxiety

 ii. Domestic violence

 iii. Sexual problems

 iv. Relationship problems

The response options were on a 4-point scale (1 = no, 2 = probably not, 3 = probably yes and 4 = yes), with higher scores indicating greater inclination to visit a primary care clinic doctor. Individuals with a score of 3 and 4 were assigned to perceived high inclination and score of 1 and 2 were assigned to perceived low inclination. There was also a "don't know" option for patients who were unable to select one of the 4-scale responses and these were treated as missing in the analysis. The prevalence of "don't know" responses were reported to illustrate the extent to which patients' were unable to express an opinion on perceived community healthcare seeking behaviour for each of the health concerns evaluated.

The covariates of interest are patients' age, sex and the type of primary care setting (public or private clinics) where the interviews were conducted. The influence of age was first explored by dividing age into quartiles and plotting the coefficients on a quartile plot to observe the effect of age on each outcome. The relationship between age and the outcome measures were shown to be non-linear and change in the estimates mainly occur between the ages of 45 and 50 years. Hence, we chose to dichotomize the age covariate to two groups: one between 18 and 49 years and another with those aged 50 years and older.

Type of primary care setting serves as a proxy measure for affordability of medical care and health financing categories. Primary care in Malaysia is mainly provided by a public sector of about 1060 health clinics with daily attendances ranging between 50 and 1000 attendances and about 7400 private general practitioner clinics, which were largely solo practices [16–18]. Healthcare in the public sector is virtually free of charge and financed almost entirely by government funds while payment in private clinics is borne out-of-pocket or via private insurance [19].

The final models were adjusted for patients' characteristics including sex, age, ethnicity, educational level, employment status, household monthly income, self-reported health status,

self-reported presence of chronic condition, whether the respondent has a family/own doctor and frequency of primary care visits in the past 6 months. Clinic-related characteristics that were included in the model were travelling time from home to clinic, waiting time between arriving at the clinic and consultation, willingness of doctor to discuss patients' dissatisfaction of treatment received and whether patients' trust doctors in general. The missing data rates for covariates ranged from 0.5% to 17.8%.

## Statistical analysis

Continuous variables were presented as mean and standard deviation while categorical variables were reported in frequencies and percentages. Chi-square test was used for univariable comparisons between groups. Multivariable logistic regression using complex survey design to account for clustering within clinics was used to estimate adjusted odds ratios (OR) and 95% confidence intervals (CI) for the association sex, age and the type of primary care setting for each of the acute and preventive physical health concerns and psychosocial health concerns. All models were adjusted for the patient and clinic-related variables described above. Multicollinearity of the covariates was checked and linearity of continuous variables was assessed using quartile plots [20]. Also, interactions between age, sex and type of primary care setting were also checked for and no significant interactions were found for each outcome. Complete case analysis was used and Forest plots were used to display the OR and 95% CI for each outcome measure. A p-value <0.05 was considered statistically significant. For the age comparison of perceived community healthcare seeking behaviour, sensitivity analyses were conducted based an age cut-off of 45 years (18–44 years and $\geq$ 45 years) and these estimates were compared with estimates for the cut-off of 50 years old in the results section. Data analyses were performed using Stata statistical software version 14.3 (StataCorp LP, College Station, TX) [21].

## Results

A total of 3979 patients completed the patient experience questionnaire. The mean age of patients was 41.9 years (SD,15.5) with more women (61.6%) and adults below the age of 50 years (67.2%) being surveyed. Table 1 showed the socio-demographic and clinic characteristics

**Table 1. Baseline characteristics of participants.**

| Characteristic | n | % |
|---|---|---|
| **Sex** | | |
| Men | 1527 | 38.4 |
| Women | 2452 | 61.6 |
| **Age (years)** | | |
| <50 | 2675 | 67.2 |
| $\geq$ 50 | 1304 | 32.8 |
| **Ethnicity** | | |
| Malay | 2082 | 52.3 |
| Chinese | 593 | 14.9 |
| Indian | 301 | 7.6 |
| Others | 1003 | 25.2 |
| **Educational level** | | |
| No formal education till lower secondary | 1249 | 31.4 |
| Upper secondary | 1537 | 38.6 |
| Post secondary and higher | 1193 | 30.0 |

(*Continued*)

**Table 1.** (Continued)

| Characteristic | n | % |
|---|---|---|
| **Household income** | | |
| < MYR5000 | 2395 | 60.2 |
| ≥ MYR5000 | 1584 | 39.8 |
| **Employment status** | | |
| Unemployed | 1379 | 34.7 |
| Employed | 2600 | 65.3 |
| **General health** | | |
| Poor | 162 | 4.1 |
| Fair | 1090 | 27.4 |
| Good | 2277 | 57.2 |
| Very good | 450 | 11.3 |
| **Longstanding condition (Yes)** | 1541 | 38.7 |
| **Have own doctor (Yes)** | 824 | 20.7 |
| **Number of GP visits in the past 6 months** | | |
| ≤3 visits | 2920 | 73.4 |
| >3 visits | 1059 | 26.6 |
| **Type of primary care setting visited** | | |
| Public | 1927 | 48.4 |
| Private | 2052 | 51.6 |
| **Location of clinic visited** | | |
| Rural | 1156 | 29.1 |
| Urban | 2823 | 70.9 |
| **Time to travel from home to practice** | | |
| ≤15minutes | 2844 | 71.5 |
| >15 minutes | 1135 | 28.5 |
| **Waiting time between arrival at practice and consultation (N = 3968)** | | |
| <15 minutes | 1615 | 40.7 |
| 15–30 minutes | 1016 | 25.6 |
| 30–45 minutes | 339 | 8.5 |
| 45–60 minutes | 237 | 6.0 |
| >60 minutes | 761 | 19.2 |
| **If you are unhappy with the treatment you received, do you think this doctor would be prepared to discuss it with you? (N = 3363) (Yes)** | 3089 | 91.9 |
| **In general, doctors can be trusted** | | |
| Strongly disagree | 4 | 0.1 |
| Disagree | 103 | 2.6 |
| Agree | 2003 | 50.3 |
| Strongly agree | 1869 | 47.0 |

If N is not stated, the total patients included for analysis was 3979.

SD, standard deviation; MYR, Malaysian Ringgit; GP, general practitioner.

of the study sample. Majority of the patients were of Malay ethnicity (52.3%), under employment (65.3%), do not have any chronic conditions (61.3%) and reported either good or very good health status (68.5%). Almost a third (30.0%) had education up to tertiary level and 60% came from the lower household income group (<MYR 5000). Nearly half of them (48.4%) were surveyed at private health clinics and about 70% of all clinics were situated in urban areas.

**Table 2. Perception of community healthcare seeking behavior to utilisation of primary care services.**

| Would most people visit a clinic doctor for: | Perceived healthcare seeking behaviorcategory | | | | | |
|---|---|---|---|---|---|---|
| | Low | | High | | Don't know | |
| | n | % | n | % | n | % |
| **Acute and preventive physical health concerns** | | | | | | |
| Stomach pain | 268 | 6.8 | 3674 | 92.3 | 37 | 0.9 |
| Blood in the stool | 312 | 7.8 | 3458 | 86.9 | 209 | 5.3 |
| Routine health checks | 385 | 9.7 | 3489 | 87.7 | 105 | 2.6 |
| Cut finger that needs to be stitched | 565 | 14.2 | 3170 | 79.7 | 244 | 6.1 |
| Deteriorated vision | 758 | 19.1 | 3042 | 76.4 | 179 | 4.5 |
| Sprained ankle | 891 | 22.4 | 2957 | 74.3 | 131 | 3.3 |
| Help to quit smoking | 1277 | 32.1 | 1993 | 50.1 | 709 | 17.8 |
| Removal of a wart | 1365 | 34.3 | 1931 | 48.5 | 683 | 17.2 |
| **Psychosocial health concerns** | | | | | | |
| Anxiety | 1466 | 36.8 | 2063 | 51.9 | 450 | 11.3 |
| Sexual problems | 1404 | 35.3 | 1909 | 48.0 | 666 | 16.7 |
| Domestic violence | 2162 | 54.3 | 1196 | 30.1 | 621 | 15.6 |
| Relationship problems | 2517 | 63.3 | 855 | 21.5 | 607 | 15.2 |

About 80% of patients claimed that they do not have their own general practitioners to first consult on a health problem. However, almost all the patients reported they trusted doctors in general and believed that the doctors they were seeing on the day of visit were prepared to discuss, should they be unhappy with the treatment received. About 30% had at least four GP visits in the last 6 months while 72% claimed they took less than 15 minutes to travel from home to the clinic. About two third (66.3%) reported the waiting time between arriving at the clinics and the consultation was within 30 minutes.

In general, the perceived community healthcare seeking behavior for primary care was higher for acute and preventive health concerns than those for psychosocial concerns, with the percentage who reported high perceived healthcare seeking behavior ranging between 48% to 93% and 21% to 52%, respectively (Table 2). Almost all the patients perceived high community utilisation of primary care services for acute and preventive physical health concerns, particularly for stomach pain (92.3%), blood in the stool (86.9%) and routine health checks (87.7%). In contrast, the perceived community primary health care utilisation was lower for psychosocial concerns, where 51.9% believe the community would consult for anxiety, followed by sexual problems (48.0%), domestic violence (30.1%) and relationship problems (21.5%). The don't know responses were lowest for stomach pain (0.9%) and highest for help to quit smoking (17.8%). The 'don't know' responses were on average more prevalent for psychosocial health concerns (14.7%) compared to acute and preventive ones (7.2%).

Table 3 showed that more women than men perceived high healthcare seeking behavior to utilisation of primary care across all health concerns surveyed. Univariable comparisons found sex differences of perceived healthcare seeking behavior were significant for health concerns such as stomach pain, blood in the stool, sprained ankle, help to quit smoking, removal of wart, anxiety, domestic violence and relationship problems. However, after adjustment for patient and clinic covariates (Fig 1), only 6 health concerns, i.e. stomach pain, sprained ankle, removal of a wart, anxiety, domestic violence and relationship problems were significantly different between men and women. Another noteworthy finding is, while perceived healthcare seeking behavior is generally higher in women, perceived healthcare seeking behavior for psychosocial health concerns were consistently lower than those for acute and preventive health

**Table 3. Perception of healthcare seeking behaviour in acute and preventive physical and psychosocial health concerns by sex.**

| Health concern | Sex | Perceived healthcare seeking behaviour | | | | p-value† |
| | | Low | | High | | |
| | | n | % | n | % | |
| **Acute and preventive physical health** | | | | | | |
| Stomach pain | Men | 123 | 8.1 | 1391 | 91.9 | 0.01 |
| | Women | 145 | 6.0 | 2283 | 94.0 | |
| Blood in the stool | Men | 141 | 9.7 | 1307 | 90.3 | 0.01 |
| | Women | 171 | 7.4 | 2151 | 92.6 | |
| Routine health checks | Men | 160 | 10.7 | 1331 | 89.3 | 0.19 |
| | Women | 225 | 9.4 | 2158 | 90.6 | |
| Cut finger that needs to be stitched | Men | 232 | 16.2 | 1202 | 83.8 | 0.16 |
| | Women | 333 | 14.5 | 1968 | 85.5 | |
| Deteriorated vision | Men | 301 | 20.7 | 1152 | 79.3 | 0.35 |
| | Women | 457 | 19.5 | 1890 | 80.5 | |
| Sprained ankle | Men | 380 | 25.6 | 1102 | 74.4 | 0.004 |
| | Women | 511 | 21.6 | 1855 | 78.4 | |
| Help to quit smoking | Men | 549 | 42.2 | 751 | 57.8 | 0.002 |
| | Women | 728 | 37.0 | 1242 | 63.0 | |
| Removal of a wart | Men | 566 | 44.2 | 716 | 55.9 | 0.01 |
| | Women | 799 | 39.7 | 1215 | 60.3 | |
| **Psychosocial health** | | | | | | |
| Anxiety | Men | 624 | 45.4 | 752 | 54.6 | <0.001 |
| | Women | 842 | 39.1 | 1311 | 60.9 | |
| Sexual problems | Men | 556 | 43.0 | 738 | 57.0 | 0.58 |
| | Women | 848 | 42.0 | 1171 | 58.0 | |
| Domestic violence | Men | 895 | 68.5 | 412 | 31.5 | <0.001 |
| | Women | 1267 | 61.8 | 784 | 38.2 | |
| Relationship problems | Men | 1023 | 77.9 | 290 | 22.1 | <0.001 |
| | Women | 1494 | 72.6 | 565 | 27.4 | |

† chi-square test between groups.

concerns across both men and women, especially pertaining to domestic violence (31.5% and 38.2%) and relationship problems (22.1% and 27.4%) (Table 3).

For the univariable analysis on age, differences were observed for routine health checks, a cut finger that needs to be stitched, deteriorated vision, removal of a wart, and sexual problems (Table 4). However, after adjusting for other patient and clinic covariates, these age differences in perceived healthcare seeking behavior becomes diminished except for the removal of warts, where older participants were more likely to perceive high healthcare seeking behavior in most people (Fig 2).

Table 5 showed that patients who visited the public clinics perceived higher healthcare seeking behavior for all acute and preventive physical and psychosocial health concerns (except stomach pain) compared to those who visited the private clinics. After adjustment for covariates, these differences in perceived healthcare seeking behavior between patients for public and private clinics remained for four acute and preventive physical health concerns and all psychosocial health concerns (Fig 3). The adjusted odds ratios with corresponding 95% CI were displayed for age, sex and type of primary care setting using Forest plots in Figs 1–3 respectively.

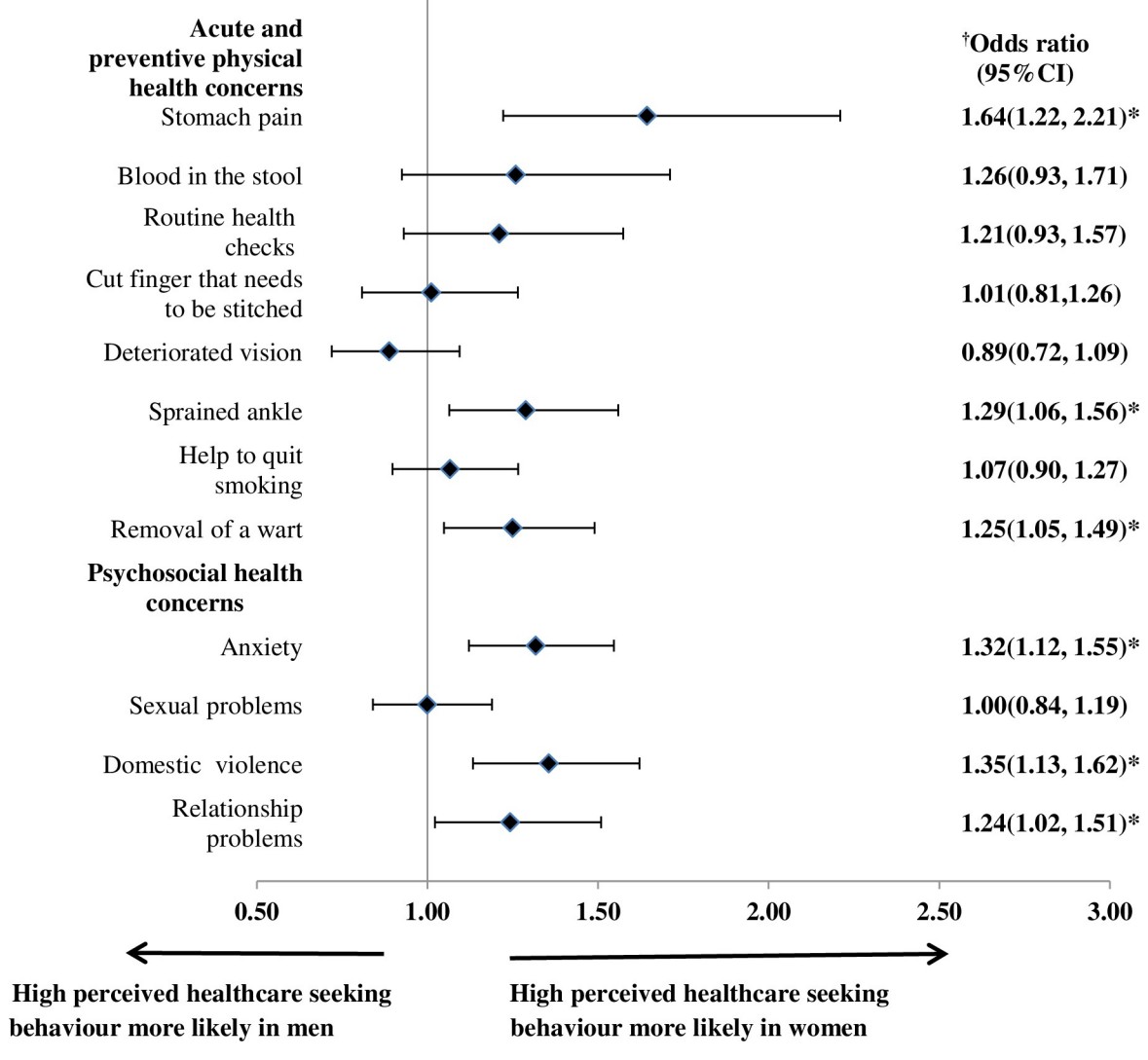

**Fig 1. Adjusted ORs with corresponding 95% CI for perceived high healthcare seeking behaviour for acute and preventive physical and psychosocial health concerns by sex.** Models were adjusted for age, ethnicity, educational level, employment status, household monthly income, self-reported general health status, self-reported presence of chronic condition, whether the patient has a family doctor and frequency of clinic visits in past 6 months, travel time from home to clinic, waiting time between arriving at the clinic and consultation, perceived willingness of the doctor to discuss if patient is unhappy with treatment received, whether patients trust doctors in general, type of primary care setting (public/private), geographical location of clinic (urban/rural). OR, odds ratio; CI, confidence interval; * statistical significant p-value <0.05.

A sensitivity analysis was done to check for differences in estimates when different age cut-offs (at 45 years and 50 years) were used for age comparisons. The results showed estimates were of similar direction and magnitude across all health concerns with the two different age cut-off points. We opted to present the results using the age categories with 50 year cut-off to allow comparisons of findings with other studies.

## Discussion

This study found that sex and type of primary care setting were significant determinants of perceived community healthcare seeking behaviour, for both acute and preventive physical health concerns and psychosocial health concerns. For health concerns which sex differences

**Table 4. Perception of healthcare seeking behaviour in acute and preventive physical and psychosocial health concerns stratified by age.**

| Health concern | Age (years) | Perceived healthcare seeking behaviour | | | | p-value† |
| --- | --- | --- | --- | --- | --- | --- |
| | | Low | | High | | |
| | | n | % | n | % | |
| **Acute and preventive physical health** | | | | | | |
| Stomach pain | < 50 | 179 | 6.7 | 2481 | 93.3 | 0.80 |
| | ≥ 50 | 89 | 6.9 | 1193 | 93.1 | |
| Blood in the stool | < 50 | 211 | 8.3 | 2344 | 91.7 | 0.95 |
| | ≥ 50 | 101 | 8.3 | 2344 | 91.7 | |
| Routine health checks | < 50 | 281 | 10.8 | 2331 | 89.2 | 0.01 |
| | ≥ 50 | 104 | 8.2 | 1158 | 91.8 | |
| Cut finger that needs to be stitched | < 50 | 415 | 16.3 | 2124 | 83.7 | 0.002 |
| | ≥ 50 | 150 | 12.5 | 1046 | 87.5 | |
| Deteriorated vision | < 50 | 564 | 22.0 | 1998 | 78.0 | <0.001 |
| | ≥ 50 | 194 | 15.7 | 1044 | 84.3 | |
| Sprained ankle | < 50 | 594 | 22.8 | 2011 | 77.2 | 0.45 |
| | ≥ 50 | 297 | 23.9 | 946 | 76.1 | |
| Help to quit smoking | < 50 | 920 | 40.0 | 1379 | 60.0 | 0.08 |
| | ≥ 50 | 357 | 36.8 | 614 | 62.2 | |
| Removal of a wart | < 50 | 981 | 43.9 | 1256 | 56.1 | <0.001 |
| | ≥ 50 | 384 | 36.3 | 675 | 63.7 | |
| **Psychosocial health** | | | | | | |
| Anxiety | < 50 | 1017 | 42.3 | 1387 | 57.7 | 0.18 |
| | ≥ 50 | 449 | 39.9 | 676 | 60.1 | |
| Sexual problems | < 50 | 926 | 40.5 | 1362 | 59.5 | 0.001 |
| | ≥ 50 | 478 | 46.6 | 547 | 53.4 | |
| Domestic violence | < 50 | 1505 | 65.3 | 800 | 34.7 | 0.10 |
| | ≥ 50 | 657 | 62.4 | 396 | 37.6 | |
| Relationship problems | < 50 | 1752 | 75.2 | 577 | 24.8 | 0.25 |
| | ≥ 50 | 765 | 73.4 | 278 | 26.6 | |

† chi-square test between groups.

were found, higher odds of perceived community healthcare seeking were consistently in one direction, i.e. women reporting higher perceived healthcare seeking behaviour than men. This finding upholds those from previous studies, where women were known to consult their general practitioners more often than men and were more proactive in health seeking [22, 23]. We found that women were more likely to perceive healthcare seeking for pain-related complaints such as stomach pain and sprained ankle and this is consistent with studies showing that women utilise health services more for pain than men [24, 25]. Possible explanations are that men could have higher threshold to pain, were less willing to report pain and could possibly have waited longer before seeking care than women [26, 27]. Besides that, we have shown that women were also more likely to perceive higher community healthcare seeking for psychosocial concerns compared to men and this confirms previous literature that women were more inclined to report on consultations for mental health issues [22, 23]. The lower likelihood of men to perceive community healthcare seeking for psychosocial concerns may be related to a lack of psychological openness among men [23]. Statistically, about a third of adults in Malaysia were shown to be facing mental health issues and prevalences did not differ between the males and females [28].Therefore, in this age where mental illness rates are rising, there is a

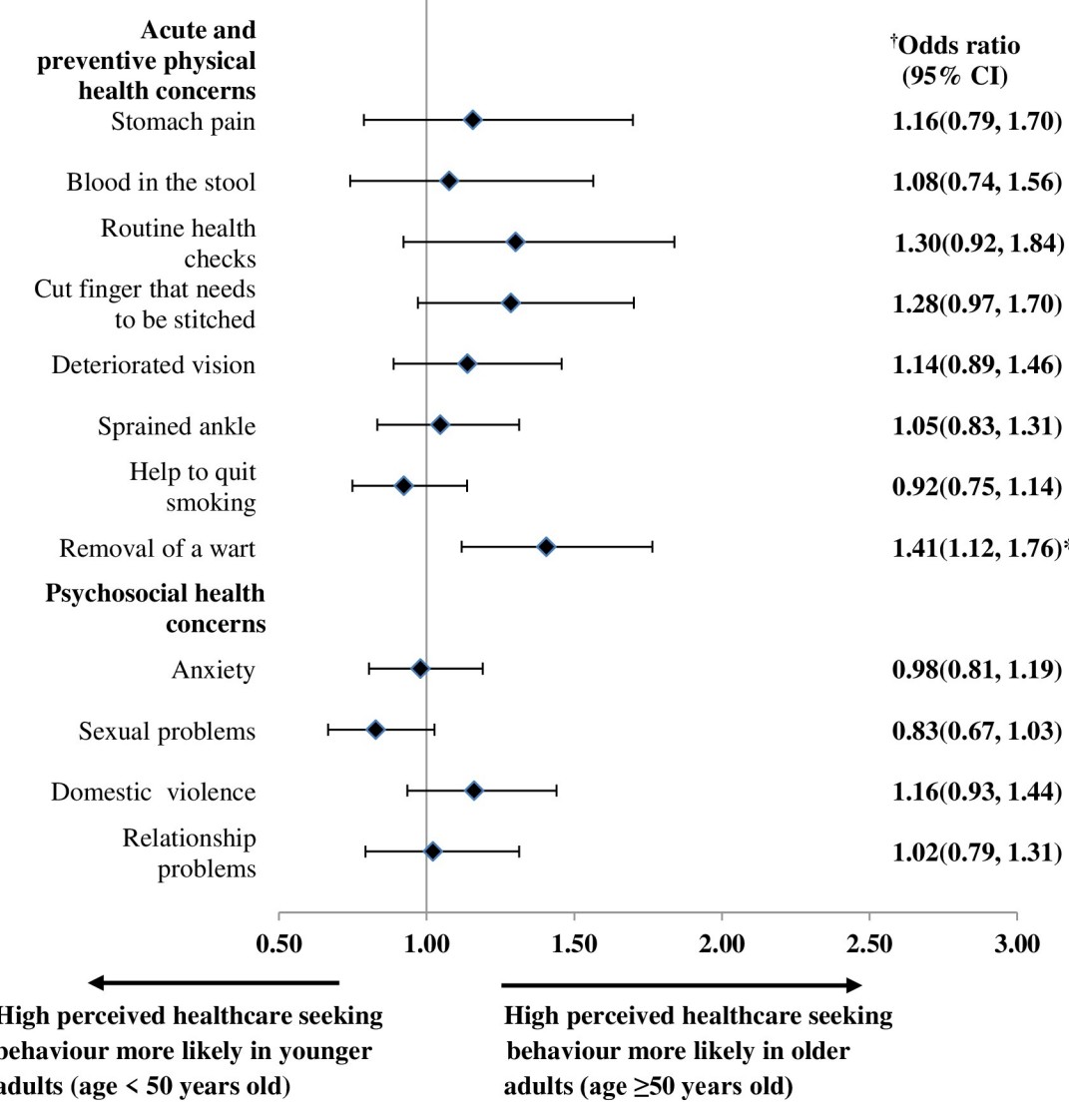

**Fig 2. Adjusted ORs with corresponding 95% CI for perceived high healthcare seeking behaviour for acute and preventive physical and psychosocial health concerns by age.** [†] Models were adjusted for sex, ethnicity, educational level, employment status, household monthly income, self-reported general health status, self-reported presence of chronic condition, whether the patient has a family doctor and frequency of clinic visits in past 6 months, travel time from home to clinic, waiting time between arriving at the clinic and consultation, perceived willingness of the doctor to discuss if patient is unhappy with treatment received, whether patients trust doctors in general, type of primary care setting (public/private), geographical location of clinic (urban/rural). OR, odds ratio; CI, confidence interval; * statistical significant p-value <0.05.

pressing need to address the low perceived healthcare seeking behaviour among men especially for psychosocial concerns.

Differences in perceived healthcare seeking tendencies were observed between those who seeked treatment at the two major types of primary care clinics in Malaysia. Compared to the private clinics, those who received care at public clinics were more likely to perceive high community healthcare seeking behaviour for eight of the 12 health concerns investigated. The primary difference between public and private clinics in Malaysia is the health financing mechanism; one that is publicly funded versus another that charges fee-for-service. Hence, the type of primary care setting serves as a proxy for differences in healthcare financing methods.

**Table 5. Perception of healthcare seeking behaviour in acute and preventive physical and psychosocial health concerns stratified by type of primary care setting.**

| Health concern | Setting | Perceived healthcare seeking behaviour | | | | p-value† |
|---|---|---|---|---|---|---|
| | | Low | | High | | |
| | | n | % | n | % | |
| **Acute and preventive physical health** | | | | | | |
| Stomach pain | Public | 127 | 6.7 | 1775 | 93.3 | 0.77 |
| | Private | 141 | 6.9 | 1899 | 93.1 | |
| Blood in the stool | Public | 98 | 5.4 | 1713 | 94.6 | <0.001 |
| | Private | 214 | 10.9 | 1745 | 89.1 | |
| Routine health checks | Public | 156 | 8.4 | 1704 | 91.6 | 0.002 |
| | Private | 229 | 11.4 | 1785 | 88.6 | |
| Cut finger that needs to be stitched | Public | 198 | 11.0 | 1603 | 89.0 | <0.001 |
| | Private | 367 | 19.0 | 1567 | 81.0 | |
| Deteriorated vision | Public | 173 | 9.4 | 1677 | 90.7 | <0.001 |
| | Private | 585 | 30.0 | 1365 | 70.0 | |
| Sprained ankle | Public | 389 | 21.2 | 1450 | 78.8 | 0.005 |
| | Private | 502 | 25.0 | 1507 | 75.0 | |
| Help to quit smoking | Public | 458 | 29.2 | 1109 | 70.8 | <0.001 |
| | Private | 819 | 48.1 | 884 | 51.9 | |
| Removal of a wart | Public | 599 | 39.1 | 932 | 60.9 | 0.013 |
| | Private | 766 | 43.4 | 999 | 56.6 | |
| **Psychosocial health** | | | | | | |
| Anxiety | Public | 606 | 36.4 | 1060 | 63.6 | <0.001 |
| | Private | 860 | 46.2 | 1003 | 53.8 | |
| Sexual problems | Public | 634 | 40.2 | 941 | 59.8 | 0.02 |
| | Private | 770 | 44.3 | 968 | 55.7 | |
| Domestic violence | Public | 941 | 59.4 | 643 | 40.6 | <0.001 |
| | Private | 1221 | 68.8 | 553 | 31.2 | |
| Relationship problems | Public | 1088 | 69.3 | 481 | 30.7 | <0.001 |
| | Private | 1429 | 79.3 | 374 | 20.7 | |

† chi-square test between group.

Higher perception of community healthcare seeking behaviour is occurring among public clinic patients. This observation is attributable to reasons such as lack of financial barriers and affordability of care, which is consistent with results by Chomi et al. where people who had health insurance were more likely to seek healthcare compared to those who were not insured [29]. This notion is also supported by another study which showed that, the lower income group gained better access to care in communities which had more government funded health facilities [30]. Another explanation for the tendency to perceive higher community healthcare seeking behaviour among public primary care attendees is the availability of a more comprehensive range of services at public clinics compared to private clinics [18].

Age was not a predictor for perceived community healthcare seeking behaviour except for the removal of warts, where people aged 50 years and above perceived higher community healthcare seeking than the lower age group. This could be partially explained by concerns among the elderly on the risks between genital warts or skin lesions and malignancies with increasing age [31, 32]. Contrary to other studies which demonstrated that older persons were less likely to use or report use of mental health services, we found no difference in perceived community healthcare seeking behaviour between age groups [22, 33]. Our results also showed

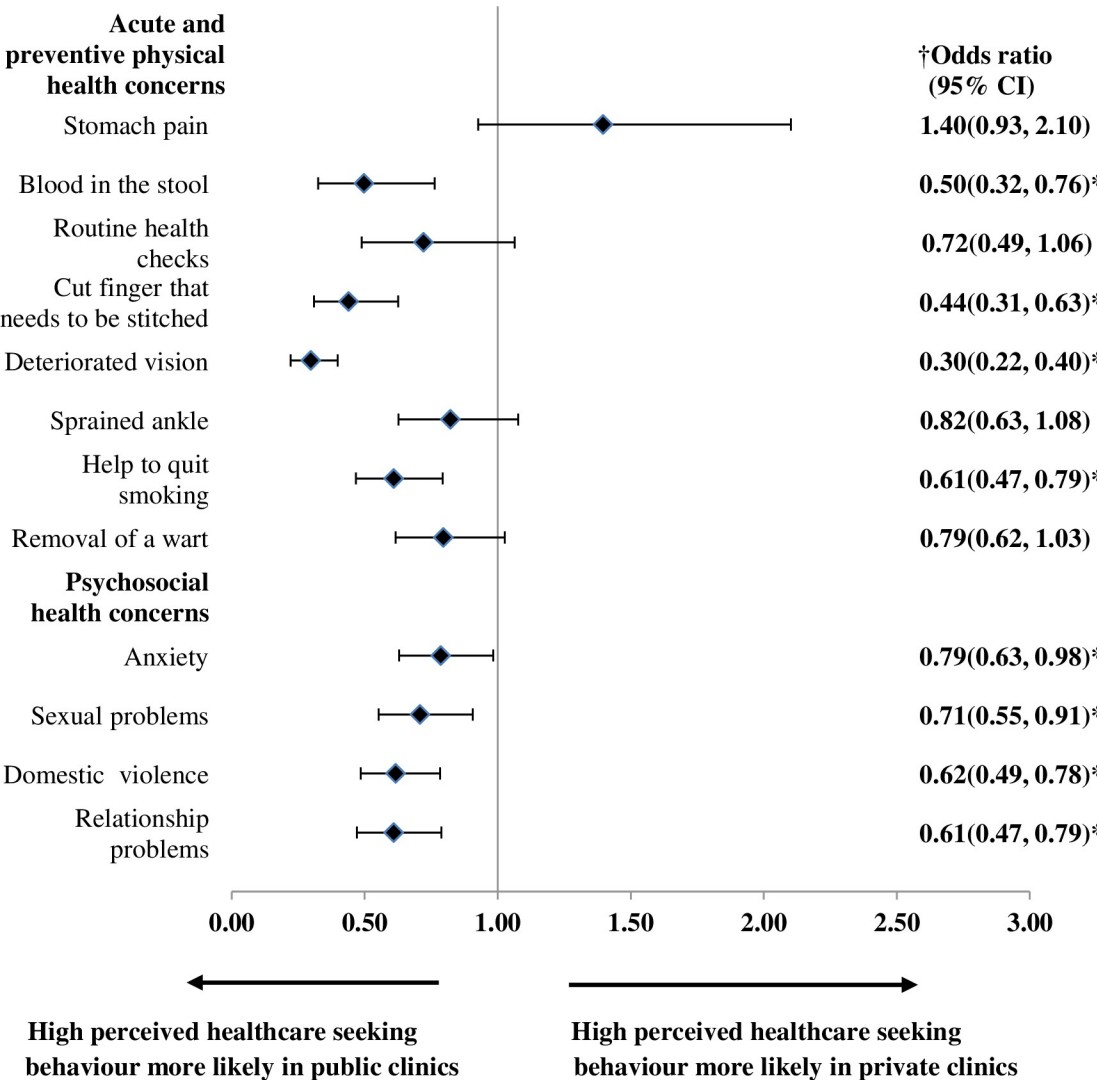

**Fig 3. Adjusted ORs with corresponding 95% CI for perceived high healthcare seeking behavior for acute and preventive physical and psychosocial health concerns by type of primary care setting.** [†] Models were adjusted for sex, age, ethnicity, educational level, employment status, household monthly income, self-reported general health status, self-reported presence of chronic condition, whether the patient has a family doctor and frequency of clinic visits in past 6 months, travel time from home to clinic, waiting time between arriving at the clinic and consultation, perceived willingness of the doctor to discuss if patient is unhappy with treatment received, whether patients trust doctors in general, geographical location of clinic (urban/rural). OR, odds ratio; CI, confidence interval; * statistical significant p-value <0.05.

no difference in perceived community health seeking for routine health examinations between the older and younger age groups and this is in contrast with findings by Deeks and colleagues, where screening behaviours were more prevalent among men and women above 50 years old [34].

We have shown that primary care clinics were not generally perceived as the first point of contact for psychosocial concerns and tobacco addiction. Unsurprisingly, primary care patients' perceived a higher tendency for the community to seek care for physical complaints than psychosocial or mental health complaints and this is in agreement with health-seeking and utilisation patterns reported by other researchers [22, 35]. However, the prevalence of perceived community health seeking for psychosocial concerns is lower compared to results from

another Malaysian study which surveyed patterns of help-seeking for common mental disorders within an urban population [36]. Similar finding was also observed in a comparative study where rural residents with mental health problems were less likely to seek help than their urban counterparts [37].

The authors did note a potential over-reporting of socially desirable behaviour because the number of subjects which reported use of complementary and alternative medicine is far too small for a setting where complementary and alternative medicine is known to play a strong influence when it comes to mental health issues [36]. Low perception of community healthcare seeking behaviour found in our study for psychosocial concerns could be attributable to belief that mental disorders were caused by supernatural occurrences and fear of stigmatization or discrimination [38]. Besides, low perception for community healthcare seeking for mental health issues at clinics may also stem from poor public awareness of the availability of mental health services at primary care clinics and insufficient knowledge and training for primary care providers to screen and manage mental disorders [39].

It is worth noting that the proportion of patients who chose the 'don't know' option was about twice higher for psychosocial issues, raising concern that the participants were either indifferent or unaware that those were potential health issues or did not know that primary care clinics were appropriate facilities to seek care from. This is in line with surveys which found low mental health literacy rates in this country [40].

This study has several strengths. While majority of studies on primary healthcare seeking behaviour have investigated utilisation patterns or individual health behaviours, few have looked at the perceived community healthcare seeking behaviour [10]. Despite difficulties in performing direct comparisons with literature, we have shown that these perceptions are broadly in line with findings on health utilisation and intention to seek primary care consultations. Another strength is the large sample size which allowed sufficient statistical power to explore sex, age and type of primary care setting differences of the study population's perspective with regard to the perception towards healthcare seeking behaviour in primary care. Also, this study had a good response rate and was conducted in a sample of clinics which are representative of both public and private primary care clinics in the country [12, 13]. One limitation of this study is that the proportion of don't knows reponses coded as missing data for the regression analysis was as high as 18%; this could potentially introduce bias to the study findings. However, this missing information could not be dealt with using methods such as imputation because the don't know responses do constitute a form of response and it is the intention of this paper to determine a specific direction for perceived healthcare seeking behaviour; i.e. whether high or low. Another limitation is that the true reason for healthcare utilisation patterns was not captured to enable triangulation of whether perceived healthcare seeking behaviour truly translates into actual utilisation.

## Implications to policy, practice and research

Steps to increase awareness of preventive and mental health services in primary care come from two main approaches; first from educating and informing the community about the availability of services and second, through direct patient engagement by primary care providers.

Despite integration of mental health services into primary care since the 1990s, the overall perceived community psychosocial health-seeking tendency is still low. Policies need to be put in place to complement the Malaysian 2012 National Mental Health Policy to actively inform and engage the community to understand the fundamentals of mental health, reduce society stigma and discrimination on mental illness and know when and where to seek help. Public

education campaigns and mental health literacy programs have been shown to be effective in reducing stigma, a main barrier to seeking health services [41]. Education on mental disorders should begin early at schools and teachers play pertinent roles in promoting mental health among children and adolescents. Another approach to promote awareness on mental health services is to provide professional, reliable and easily accessible health resources on the web makes full use of the broad reach of the internet and counters misinformation that circulate on social media [41].

Perceived community healthcare seeking behaviour for preventive services in primary care were high in general and similar between sexes and age groups. The major difference noted was in the lower perceived community healthcare seeking tendency in men for psychosocial health services. Hence, there is a need to emphasize public education on engaging men and encouraging them to come forward to seek help for mental health concerns at primary care settings as it was revealed in the Asian Men's Health Report that there was a higher mortality rate due to suicide for men compared to women [42]. Screening rate for mental illness are being conducted by public primary care clinics but the screening rates are still below 10% [43]. Among the reasons to this are the high workload in public clinic which limits meaningful interactions between provider and patient as well as the lack of confidence in providers to detect and manage mental disorders. Therefore, it is very important to strengthen the clinical skills for early detection, diagnosis, management and counselling among primary care providers through formal training at both the undergraduate and postgraduate level [36]. Further, it is equally important for private general practitioners to be equipped with the skills to engage patients when it comes to promoting screening for mental disorders in primary care. It is also highly important that implementation of all policy and practice changes are scientifically and systematically evaluated through research to determine effectiveness of these efforts.

## Conclusions

We found that there were sex and type of primary care setting differences when it comes to acute and preventive physical health concerns as well as psychosocial health concerns. Differences in perceived healthcare seeking tendencies were however, not observed between those aged below 50 years and 50 years and above. Overall, perceived community healthcare seeking for primary care services was low for psychosocial or mental health concerns and there was still a substantial proportion of patients who were unaware that they could seek primary care services for mental health complaints.

## Acknowledgments

We wish to thank the Director-General of Health, Ministry of Health Malaysia for permission to publish the findings. We would also like to thank the patients and health providers who participated in the QUALICOPC study and the QUALICOPC study investigators. We wish to acknowledge the contribution and support of Professor Rifat Atun from Harvard T.H. Chan School of Public Health, Harvard University and Willemijn Schäfer, Wienke Boerma and Peter Groenewegen from Netherlands Institute for Health Services Research (NIVEL) in the planning and conduct of the Malaysian QUALICOPC study.

## Author Contributions

**Conceptualization:** Ming Tsuey Lim, Yvonne Mei Fong Lim, Seng Fah Tong, Sheamini Sivasampu.

**Formal analysis:** Ming Tsuey Lim, Yvonne Mei Fong Lim.

**Funding acquisition:** Sheamini Sivasampu.

**Methodology:** Ming Tsuey Lim, Yvonne Mei Fong Lim, Seng Fah Tong, Sheamini Sivasampu.

**Project administration:** Sheamini Sivasampu.

**Visualization:** Ming Tsuey Lim.

**Writing – original draft:** Ming Tsuey Lim, Yvonne Mei Fong Lim.

**Writing – review & editing:** Ming Tsuey Lim, Yvonne Mei Fong Lim, Seng Fah Tong, Sheamini Sivasampu.

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
