## [Decision Letter · Decision Letter 0]

23 Aug 2019

PONE-D-19-17088

Age, sex and primary care setting differences in patients’ perception of community healthcare seeking behaviour to health services

PLOS ONE

Dear Ms Lim,

Thank you for submitting your manuscript to PLOS ONE. After careful consideration, we feel that it has merit but does not fully meet PLOS ONE’s publication criteria as it currently stands. Therefore, we invite you to submit a revised version of the manuscript that addresses the points raised during the review process.

We would appreciate receiving your revised manuscript by Oct 07 2019 11:59PM. To enhance the reproducibility of your results, we recommend that if applicable you deposit your laboratory protocols in protocols.io, where a protocol can be assigned its own identifier (DOI) such that it can be cited independently in the future. For instructions see: http://journals.plos.org/plosone/s/submission-guidelines#loc-laboratory-protocols

We look forward to receiving your revised manuscript.

Kind regards,

Wen-Jun Tu

Academic Editor

PLOS ONE

Journal Requirements:

Additional Editor Comments (if provided):

Reviewers' comments:

Reviewer's Responses to Questions

**Comments to the Author**

1. Is the manuscript technically sound, and do the data support the conclusions?

Reviewer #1: No

Reviewer #2: Yes

Reviewer #3: Yes

2. Has the statistical analysis been performed appropriately and rigorously? 

Reviewer #1: Yes

Reviewer #2: Yes

Reviewer #3: Yes

3. Have the authors made all data underlying the findings in their manuscript fully available?

Reviewer #1: No

Reviewer #2: Yes

Reviewer #3: No

4. Is the manuscript presented in an intelligible fashion and written in standard English?

Reviewer #1: No

Reviewer #2: Yes

Reviewer #3: Yes

5. Review Comments to the Author

Reviewer #1: The scientific value of this manuscript is questionable and I don't think that it will have positive impact in the field

Reviewer #2: Dear Authors,

The publication is a good one with much variability's taken and can be accepted from my point of view, but there are some points which must be taken into consideration:

1. Try to avoid repetition of the same sentences as in the abstract (Healthcare seeking ...) within the same paragraph.

2. In some paragraphs you left spaces at the beginning in others not. Please unify

3. In the tables, they are very long. It is advisable to separate them each for a criteria, so that you would have precise characters within one table to avoid long tables for the reader.

Reviewer #3: Congratulations to the authors for this interesting manuscript regarding the perceived community healthcare seeking behavior for acute and preventive physical and psychosocial health concerns in Malaysian primary care clinics. The authors took into consideration sex, age and affordability of care (health insurance) as key elements of healthcare seeking behaviors in Malaysian community. This is a continuation of the efforts in the previous publications, Quality and Costs of Primary Care(QUALICOPC) study in Malaysia: Phase I –Public Clinics, and Quality and Costs of Primary Care (QUALICOPC) Malaysia: Phase II – Private Clinics (cited as references 12, 13) in understanding potential determinants of community healthcare seeking behaviors in Malaysia.

Comments:

1) Line 116: The data for this study was extracted from the International Quality and Cost of Primary Care (QUALICOPC) study: Can you please add a citation for this statement? you may cite the data used in the study if previously published, alongside with references 12 and 13.

2) Lines 124-125: Further details on the methods of this study are described elsewhere: can you please explain more what these details are for the reader? Did you mean sampling frame, sampling methods, and eligibility of participants?

3) Lines 128-134: Ten patients aged 18 years old and above from each clinic were invited to participate in the survey:

The sample size is 3979 patients. Total number of clinics is 221+239= 460 clinics.

Overall response rate for both patient questionnaires was 91.1%.

One patient will be administered the patient values questionnaire while nine will be administered the patient experience questionnaire.

Since the overall sample size was 3979, can you please clarify if there were any excluded patients for incomplete data or because of the exclusion criteria?

4) Lines 129-130: the patient values questionnaire and patient experience questionnaire were mentioned before the appearance of reference 15 in Survey Tool part (which can be considered as a reference for them too).

5) 337-339: please add references to this statement (you can use one of those mentioned later on in this paragraph).

6) Lines 385- 388: A comparison between urban and rural clinics visitors in seeking medical attention for psychosocial or mental health complaints, in addition to acute and preventive physical health concerns, can be considered.

7) Some minor writing errors within the text are present. It is suggested to carefully review the manuscript towards spelling and/or grammatical errors. Some examples are given below:

Lines 55-56: lower than for physical health concerns.

Line 92: [8] while > [8], while

Line 145: were presented > was presented (since the verb is for the sub-section, not the 12 items)

Line 225: they do not their have own general practitioners > do not have their own

Line 406: behaviour [10. > behaviour [10].

6. PLOS authors have the option to publish the peer review history of their article (what does this mean?). If published, this will include your full peer review and any attached files.

Reviewer #1: No

Reviewer #2: Yes: Adnan Al Lahham

Reviewer #3: Yes: Isam Bsisu

---

## [Author Response · Author response to Decision Letter 0]

11 Sep 2019

Responses to comments from Reviewer #1

Have the authors made all data underlying the findings in their manuscript fully available? 

Q1) No 

R1) Yes, we have consulted the principal investigator of the data source and agreed to include our de-identified dataset with relevant variables. As such our data availability statement will be amended to

All relevant data are within the manuscript and its Supporting Information files.

Review Comments to the Author

Q2) The scientific value of this manuscript is questionable and I don't think that it will have positive impact in the field

R2) We appreciate the reviewer’s concern.

We would like to reassure the reviewer that our findings in this paper would help in the understanding of healthcare utilisation pattern by type of health concerns developing of healthcare policy and planning for effective treatment and appropriate interventions for management of health concerns. Furthermore, early recognition presentation to health care facilities and compliance with effective treatment have shown to reduce morbidity and thereby mortality [1-3].

1. Van der Hoeven M, Kruger A, Greeff M. Differences in health care seeking behaviour between rural and urban communities in South Africa. International Journal for Equity in Health 2012; 11: 31.

2. Hausmann-Mueala S, Muela Ribera J, Nyamongo I: Health-seeking behaviour and the health system response. Disease Control Priorities Project (DCPP). Working Paper no.14;. 2003

3. World Health Organization: Rapid assessment of health seeking behaviour in relation to sexual transmitted disease: draft protocol.1995

Responses to comments from Reviewer #2

Review Comments to the Author

The publication is a good one with much variability's taken and can be accepted from my point of view, but there are some points which must be taken into consideration:

Q1) Try to avoid repetition of the same sentences as in the abstract (Healthcare seeking ...) within the same paragraph.

R1) We thank the reviewer for his comments and suggestions. 

We rephrased the following paragraph to avoid repetitions of same sentence as in the introduction section which read previously

“Healthcare seeking behaviour is also influenced by various determinants such as sex, age, social status, type of illness, access to services and perceived quality of the service [1, 2]. Understanding these potential determinants of healthcare seeking behaviour will be necessary in improving healthcare utilisation and health outcomes within different populations.”

 to

Introduction , page 4, line 78-82

Studies have shown that various determinants such as sex, age, social status, type of illness, access to services and perceived quality of the service [1, 2] affect an individual’s healthcare seeking behaviour but findings have been inconsistent. As such, it is necessary to understand these potential determinants in improving healthcare utilisation and health outcomes within different populations.

Q2) In some paragraphs you left spaces at the beginning in others not. Please unify

R2) We have removed the space at the beginning of every paragraph. To differentiate between paragraphs, we inserted an empty line between paragraphs.

Q3) In the tables, they are very long. It is advisable to separate them each for a criteria, so that you would have precise characters within one table to avoid long tables for the reader.

R3) We acknowledge the concerns raised by the reviewer. However, we believe that the characteristics listed within Table 1 would all complement each other to provide baseline characteristics of our study participants. 

We agree that the table is rather long and have amended it. We rephrased the question into more concise phrase. 

Kindly refer to Table 1 within the manuscript.

Results, page 11-13, line 247-249 

Response to comments from Reviewer #3

Have the authors made all data underlying the findings in their manuscript fully available?

Q1) No

R1) Yes, we have consulted the principal investigator of the data source and agreed to include our de-identified dataset with relevant variables. As such our data availability statement will be amended to 

All relevant data are within the manuscript and its Supporting Information files.

Review Comments to the Author

Congratulations to the authors for this interesting manuscript regarding the perceived community healthcare seeking behavior for acute and preventive physical and psychosocial health concerns in Malaysian primary care clinics. The authors took into consideration sex, age and affordability of care (health insurance) as key elements of healthcare seeking behaviors in Malaysian community. This is a continuation of the efforts in the previous publications, Quality and Costs of Primary Care(QUALICOPC) study in Malaysia: Phase I –Public Clinics, and Quality and Costs of Primary Care (QUALICOPC) Malaysia: Phase II – Private Clinics (cited as references 12, 13) in understanding potential determinants of community healthcare seeking behaviors in Malaysia.

Comments:

Q1) Line 116: The data for this study was extracted from the International Quality and Cost of Primary Care (QUALICOPC) study: Can you please add a citation for this statement? you may cite the data used in the study if previously published, alongside with references 12 and 13.

R1) We thank the reviewer for his comments and suggestions. 

We have added the references as follows:

Materials and methods, page 5, line 118-119.

The data for this study was extracted from the International Quality and Cost of Primary Care (QUALICOPC) study conducted in the primary care setting in Malaysia [12, 13].

Q2) Lines 124-125: Further details on the methods of this study are described elsewhere: can you please explain more what these details are for the reader? Did you mean sampling frame, sampling methods, and eligibility of participants?

R2) Yes, and we have amended the sentence. It now reads: 

Materials and methods, page 6, line 127-129

Further details on the methods including study design, sampling methods, questionnaires and eligibility criteria are described elsewhere [12, 13].

Q3) Lines 128-134: Ten patients aged 18 years old and above from each clinic were invited to participate in the survey:

The sample size is 3979 patients. Total number of clinics is 221+239= 460 clinics.

Overall response rate for both patient questionnaires was 91.1%.

One patient will be administered the patient values questionnaire while nine will be administered the patient experience questionnaire.

Since the overall sample size was 3979, can you please clarify if there were any excluded patients for incomplete data or because of the exclusion criteria?

R3) We added the following statements and hope that it would provide better clarity on the number of respondents. 

Materials and methods, page 6, line 137-141

A total of 4983 patients were invited and the overall response rate for both patient questionnaires was 91.1%. A non-response analysis was not necessary as it is usually conducted only when response rates fall below 80% [15]. One hundred and three respondents were excluded because of incomplete data. Of the two types of questionnaires administered, we used data from only the patient experience survey in this analysis.

Q4) Lines 129-130: the patient values questionnaire and patient experience questionnaire were mentioned before the appearance of reference 15 in Survey Tool part (which can be considered as a reference for them too).

R4) We inserted the required reference as suggested as follows:

Materials and methods, page 6, line 133-134

One patient will be administered the patient values questionnaire while nine will be administered the patient experience questionnaire [14].

Q5) 337-339: please add references to this statement (you can use one of those mentioned later on in this paragraph).

R5) We inserted the required references as suggested as follows:

Discussion, page 22, line 350-352

This finding upholds those from previous studies, where women were known to consult their general practitioners more often than men and were more proactive in health seeking [22-23].

22) Thompson AE, Anisimowicz Y, Miedema B,Hogg W, Wodchis WP, Bassler KA. The influence of gender and other patient characteristics on health care-seeking behaviour: a QUALICOPC study. BMC Family Practice 2016; 17: 38.

23) Mackenzie CS, Gekoski WL, Knox VJ. Age, gender, and the underutilization of mental health services: the influence of help-seeking attitudes. Aging Ment Health 2006; 10: 574–582.

Q6) Lines 385- 388: A comparison between urban and rural clinics visitors in seeking medical attention for psychosocial or mental health complaints, in addition to acute and preventive physical health concerns, can be considered.

R6) We included a comparative study to support our finding on help-seeking behaviour for mental health problem between rural and urban population.

Discussion, page 24, line 401-406

However, the prevalence of perceived community health seeking for psychosocial concerns is lower compared to results from another Malaysian study which surveyed patterns of help-seeking for common mental disorders within an urban population [36]. Similar finding was also observed in a comparative study where rural residents with mental health problems were less likely to seek help than their urban counterparts [37].

36) Ismail SIF. Patterns and risk factors with help- seeking for common mental disorders in an urban Malaysian community. London School of Hygiene and Tropical Medicine, 2011

37) Caldwell TM, Jorm AF, Dear KBG. Suicide and mental health in rural, remote and metropolitan areas in Australia. The Medical Journal of Australia 2004; 181: S10.

Q7) Some minor writing errors within the text are present. It is suggested to carefully review the manuscript towards spelling and/or grammatical errors. Some examples are given below:

R7) Thank you for highlighting the spelling and/or grammatical errors. We have corrected those mistakes as follow:

Q7a) Lines 55-56: lower than for physical health concerns.

R7a) Abstract, page 3, line 53-56

Our findings showed that sex and healthcare affordability differences were present in perceived community healthcare seeking behaviour towards primary care services. Also, perceived healthcare seeking behaviour was consistently lower for psychosocial health concerns compared to physical health concerns.

Q7b) Line 92: [8] while > [8], while

R7b) Introduction, page 4, line 92-94

Thus far, patient surveys on healthcare seeking behaviour have often focused on disease specific areas such as tuberculosis [7] or depression [8], while less attention has been given to primary care in general.

Q7c) Line 145: were presented > was presented (since the verb is for the sub-section, not the 12 items)

R7c) Introduction, page 7, line 152-154

Only the sub-section on the community healthcare seeking behaviour perception (12 items) from the patient experience questionnaire was presented in this analysis and all patients who responded to this questionnaire were included. 

Q7d) Line 225: they do not their have own general practitioners > do not have their own

R7d) Results, page 10, line 239-240

About 80% of patients claimed that they do not have their own general practitioners to first consult on a health problem. 

Q7e) Line 406: behaviour [10. > behaviour [10].

R7e) Discussion, page 25, line 424-426

While majority of studies on primary healthcare seeking behaviour have investigated utilisation patterns or individual health behaviours, few have looked at the perceived community healthcare seeking behaviour [10].

Additional information (From Ms Michelle Ellis)

Q1) Please include a copy of the interview guide used in the study, in both the original language and English, as Supporting Information, or include a citation if it has been published previously

R1) The copy of patient questionnaires in English, Malay and Mandarin languages are available at the citation as mentioned in following section of the manuscript

Materials and methods, page 6, line 127-129

Further details on the methods including study design, sampling methods, questionnaires and eligibility criteria are described elsewhere [12, 13].

---

## [Decision Letter · Decision Letter 1]

23 Sep 2019

PONE-D-19-17088R1

Age, sex and primary care setting differences in patients’ perception of community healthcare seeking behaviour towards health services

PLOS ONE

Dear Ms Lim,

Thank you for submitting your manuscript to PLOS ONE. After careful consideration, we feel that it has merit but does not fully meet PLOS ONE’s publication criteria as it currently stands. Therefore, we invite you to submit a revised version of the manuscript that addresses the points raised during the review process.

We would appreciate receiving your revised manuscript by Nov 07 2019 11:59PM. To enhance the reproducibility of your results, we recommend that if applicable you deposit your laboratory protocols in protocols.io, where a protocol can be assigned its own identifier (DOI) such that it can be cited independently in the future. For instructions see: http://journals.plos.org/plosone/s/submission-guidelines#loc-laboratory-protocols

We look forward to receiving your revised manuscript.

Kind regards,

Wen-Jun Tu

Academic Editor

PLOS ONE

Additional Editor Comments (if provided):

In order to provide a more complete information to our readers on the topic, we would like to emphasize the importance to cross referencing very recent material on the same topic published in "PLoS ONE ". Therefore, it would be highly appreciated if you would check the contents published in the last two years of "PLoS ONE" (https://journals.plos.org/plosone/) and add all material relevant to your article to the reference list.

Reviewers' comments:

Reviewer's Responses to Questions

**Comments to the Author**

1. If the authors have adequately addressed your comments raised in a previous round of review and you feel that this manuscript is now acceptable for publication, you may indicate that here to bypass the “Comments to the Author” section, enter your conflict of interest statement in the “Confidential to Editor” section, and submit your "Accept" recommendation.

Reviewer #1: (No Response)

Reviewer #2: All comments have been addressed

Reviewer #3: All comments have been addressed

2. Is the manuscript technically sound, and do the data support the conclusions?

Reviewer #1: No

Reviewer #2: Yes

Reviewer #3: Yes

3. Has the statistical analysis been performed appropriately and rigorously? 

Reviewer #1: Yes

Reviewer #2: Yes

Reviewer #3: Yes

4. Have the authors made all data underlying the findings in their manuscript fully available?

Reviewer #1: Yes

Reviewer #2: Yes

Reviewer #3: Yes

5. Is the manuscript presented in an intelligible fashion and written in standard English?

Reviewer #1: Yes

Reviewer #2: Yes

Reviewer #3: Yes

6. Review Comments to the Author

Reviewer #1: I want to thank the authors for their effort in this manuscript entitled "Age, sex and primary care setting differences in patients’ perception of community healthcare seeking behaviour towards health services'.

With all my respect to the authors, again the scientific value of this manuscript is questionable and I don't think that it will have a positive impact in the field.

My comments:

First: I think the sub-section (12 items) of the main tool of QUALICOPC only can’t measure the community health seeking behavior.

Second: The main question is “Would most people visit a clinic doctor for?”. In my opinion, the question is asking the patients what other people expected to do. I can talk for myself, not about what others think.

Reviewer #2: I have read carefully the authors answers to my points of view regarding the manuscript. I do accept his or their points. The current paper satisfies my review points and can be accepted in this form

Reviewer #3: Reviewer comments have been adequately addressed and the manuscript adapted accordingly. I would like to take this opportunity to congratulate the authors for their interesting article, and encourage them to work on increasing the awareness of preventive and mental health services in primary care through educating and informing the community about the availability of these services, in addition to direct patient engagement by primary care provider, as recommended by their article.

7. PLOS authors have the option to publish the peer review history of their article (what does this mean?). If published, this will include your full peer review and any attached files.

Reviewer #1: No

Reviewer #2: Yes: Adnan Al Lahham

Reviewer #3: Yes: Isam Bsisu

---

## [Author Response · Author response to Decision Letter 1]

25 Sep 2019

Responses to comments from Reviewer #1

Q1) I want to thank the authors for their effort in this manuscript entitled "Age, sex and primary care setting differences in patients’ perception of community healthcare seeking behaviour towards health services'.

With all my respect to the authors, again the scientific value of this manuscript is questionable and I don't think that it will have a positive impact in the field.

My comments:

First: I think the sub-section (12 items) of the main tool of QUALICOPC only can’t measure the community health seeking behavior.

Second: The main question is “Would most people visit a clinic doctor for?”. In my opinion, the question is asking the patients what other people expected to do. I can talk for myself, not about what others think.

R1) We appreciate the reviewer’s comments. We agree that evaluating patient’s health seeking would necessitate a direct evaluation of the patient. We also agree with the reviewer that we were not directly measuring the community healthcare seeking behaviour. Nevertheless, we are taking a different angle. We apologise for not being sufficiently clear enough. We were evaluating our primary health care responsiveness to community needs. Thus, the focus is on community perception rather than patient direct health seeking behaviour. 

The objective of the survey question for the perceived health seeking behaviour is regarding the perceptions of patients about conditions warranting a visit to our primary care services. As such, the main question is designed as “Would most people visit a clinic doctor for?”. This refers to what a person perceives the community would seek in terms of healthcare.

Therefore, we have revised some of sentences in the introduction section of our manuscript and hope that it would provide better clarity on our objective which is to describe community perception of healthcare seeking behaviour.

Introduction, page 4, line 88-90 

In this study, we focus on perceived community healthcare seeking behaviour in primary care as primary care is often patients’ first point of contact with the healthcare system for most medical problems.

Introduction, page 4-5, Line 94-101:

Understanding this is important because the community perception provides another indicator of primary care utilisation. We aim to bridge this gap in knowledge on healthcare seeking behaviour which will not only reflect the patterns of perceived healthcare seeking behaviour at the different stages of utilisation, access and barriers to healthcare services in a given community but also the potential determinants to community perceptions. [9]. Hence, perceived healthcare seeking behaviour can serve as a tool to understand how healthcare services are used and also to gauge future demand for healthcare services.

We believe this patients’ perception of community of healthcare seeking behaviour study is important as it reflects the ‘Comprehensiveness of services’ where patients’ views on the breadth of the clinical task profile of services offered by the GP are taken into consideration.

Our findings of this study would hopefully have direct implications to practice and healthcare policy maker such as efforts to increase awareness of available healthcare services especially in psychosocial health concerns should be made to address this gap in healthcare, to improve barriers to health care access by identifying those who are more likely to engage in health care-seeking behaviours and the variables predicting health care-seeking and consequently, those who are not accessing primary care can be targeted and policies can be developed and put in place to promote their health care-seeking behaviour.

In addition, previous studies have demonstrated that information gathered on health care seeking behaviour would not only help in the understanding of healthcare utilisation pattern by type of health concerns but also crucial for developing health care policies and planning for early diagnosis, effective treatment and appropriate interventions [1-3]. 

References

1.Van der Hoeven M, Kruger A, Greeff M. Differences in health care seeking behaviour between rural and urban communities in South Africa. International Journal for Equity in Health 2012; 11: 31.

2.Hausmann-Mueala S, Muela Ribera J, Nyamongo I: Health-seeking behaviour and the health system response. Disease Control Priorities Project (DCPP). Working Paper no.14; 2003.

3.World Health Organization: Rapid assessment of health seeking behaviour in relation to sexual transmitted disease: draft protocol 1995.

---

## [Editor Report · Decision Letter 2]

10 Oct 2019

Age, sex and primary care setting differences in patients’ perception of community healthcare seeking behaviour towards health services

PONE-D-19-17088R2

Dear Dr. Lim,

We are pleased to inform you that your manuscript has been judged scientifically suitable for publication and will be formally accepted for publication once it complies with all outstanding technical requirements.

With kind regards,

Wen-Jun Tu

Academic Editor

PLOS ONE
---

## [Editor Report · Acceptance letter]

14 Oct 2019

PONE-D-19-17088R2 

Age, sex and primary care setting differences in patients’ perception of community healthcare seeking behaviour towards health services 

Dear Dr. Lim:

I am pleased to inform you that your manuscript has been deemed suitable for publication in PLOS ONE. Congratulations! Your manuscript is now with our production department. 

With kind regards,

on behalf of

Dr. Wen-Jun Tu 

Academic Editor

PLOS ONE